# Automated Generation of Synoptic Reports from Narrative Pathology Reports in University Malaya Medical Centre Using Natural Language Processing

**DOI:** 10.3390/diagnostics12040879

**Published:** 2022-04-01

**Authors:** Wee-Ming Tan, Kean-Hooi Teoh, Mogana Darshini Ganggayah, Nur Aishah Taib, Hana Salwani Zaini, Sarinder Kaur Dhillon

**Affiliations:** 1Data Science & Bioinformatics Laboratory, Institute of Biological Sciences, Faculty of Science, University of Malaya, Kuala Lumpur 50603, Malaysia; tanwmg@gmail.com (W.-M.T.); mdganggayah@gmail.com (M.D.G.); 2Laboratory Department, Sunway Medical Centre, Bandar Sunway 47500, Malaysia; kean75@hotmail.com; 3Department of Surgery, Faculty of Medicine, University of Malaya, Kuala Lumpur 50603, Malaysia; naisha@um.edu.my; 4Department of Information Technology, University Malaya Medical Centre, Kuala Lumpur 50603, Malaysia; hana@ummc.edu.my

**Keywords:** pathology reporting, synoptic reporting, information extraction, text mining, natural language processing, rule based

## Abstract

Pathology reports represent a primary source of information for cancer registries. University Malaya Medical Centre (UMMC) is a tertiary hospital responsible for training pathologists; thus narrative reporting becomes important. However, the unstructured free-text reports made the information extraction process tedious for clinical audits and data analysis-related research. This study aims to develop an automated natural language processing (NLP) algorithm to summarize the existing narrative breast pathology report from UMMC to a narrower structured synoptic pathology report with a checklist-style report template to ease the creation of pathology reports. The development of the rule-based NLP algorithm was based on the R programming language by using 593 pathology specimens from 174 patients provided by the Department of Pathology, UMMC. The pathologist provides specific keywords for data elements to define the semantic rules of the NLP. The system was evaluated by calculating the precision, recall, and F1-score. The proposed NLP algorithm achieved a micro-F1 score of 99.50% and a macro-F1 score of 98.97% on 178 specimens with 25 data elements. This achievement correlated to clinicians’ needs, which could improve communication between pathologists and clinicians. The study presented here is significant, as structured data is easily minable and could generate important insights.

## 1. Introduction

In the year 2020, there were 48,639 Malaysians diagnosed and reported with cancer. Among them, 17.3% (8418) had suffered from breast cancer [1]. Each verified cancer diagnosis is based on tissue histology, which is documented in a pathology report. A breast pathology report is a medical document that contains the description of breast cells and tissues, called specimens, made by a pathologist based on microscopic evidence and used to make a diagnosis of disease [2]. By reading the description from the report, the clinicians can determine whether the tissue is cancerous or noncancerous, and consequently decide the best treatment solution for the patient.

To date, traditional narrative pathology reporting comprises of the following three main sections: Macroscopy, microscopy, and gross description, which is still the preferable standard in the most clinical institutions [3,4,5], especially in University Malaya Medical Centre (UMMC), which served as the training centre to train the trainees on how to write a pathology report. These reports represent a rich source of information on detailed tumor characteristics. Unfortunately, this traditional reporting in the free-text format is usually associated with complex explanations and heterogeneous and inconsistent terminologies in describing tumors. Hence, extracting the desired information, such as breast laterality involved or histology from pathology reports, must be done manually by clinicians who can understand the contents of these reports [6]. This process can be time consuming, thereby limiting the ability of cancer registries to identify the key data elements with their response from existing pathology reports and restricting time-sensitive applications such as precision medicine [7].

In contrast to traditional surgical pathology reporting, synoptic reporting is a process for reporting data in a simplified manner in surgical pathology reports [4]. In the past decades, the College of American Pathologists (CAP) has been publishing the most comprehensive set of synoptic cancer protocols in promoting synoptic reporting in the clinical domain [8]. Synoptic reporting has advantages for a variety of stakeholders of surgical pathology reports. For pathologists, it improves the completeness, accuracy, and ease of creating the pathology report by using the checklist-style reporting format [3,9,10]. For clinicians, synoptic reports reduce errors and increase efficiency when extracting data from the pathology reports [4,11]. Moreover, it helps researchers ensure the data are amenable to scalable data capture, interoperability, and exchange, thus enabling the creation of structured datasets to facilitate the research process [12].

Despite the fact that synoptic reporting increases the quality of reports, manually converting existing narrative reports to standardize the data can be costly and time consuming. Automated conversion of the narrative pathology report to synoptic format is an active area of research that utilizes artificial intelligence (AI) to identify key data elements from narrative pathology reports, thereby reducing human efforts. Recently, the medical domains have been adopting natural language processing (NLP) in medical tasks, particularly in extracting specific information from electronic medical records (EMRs) with promised high accuracy from 85% to 98%. Some prominent examples are found in the classification of hip fracture [13], detection of thromboembolic diseases [14], and extracting actionable findings of appendicitis [15]. Despite that, most NLP techniques mainly address the primary task. In addition, NLP has been applied in the telehealth system. For example, the Covenant University Doctor (CUDoctor) with a system usability scale score of 80.4 provides health diagnosis services based on fuzzy logic rules and fuzzy inference to predict the disease based on the symptoms inputted by the end-user [16]. The NLP identifies the user-inputted symptoms before being forwarded to the decision-making system. Other than using text, the Zenbo Project had developed a service robot that enables the human–robot interactions to provide consultation to the patient based on speech [17]. NLP plays a critical role in extracting, sorting, and converting text into data that suit the deep learning model. Another example of utilizing NLP in the healthcare domain to minimize human intervention is the use of the REST service, which provides automatic authorization for healthcare services with a correctness of classification of 95.54% [18]. The REST service combines image processing and NLP to perform information extraction from the scanned medical prescription and assess the authorization of medical prescriptions.

In this study, our goal is to develop an automated NLP algorithm to summarize the existing narrative breast pathology report from UMMC to a structured synoptic pathology report along with the named entity. The structured data help clinicians to process data faster and effortlessly. Moreover, they provide a checklist-style report template to ease the creation of pathology reports in the future.

## 2. Materials and Methods

### 2.1. Study Setting

The study was conducted at the Department of Pathology, UMMC. A total of 298 narrative breast pathology reports with 593 specimens in Docx format from 174 patients written by UMMC pathologists were obtained from the Laboratory Information System (LIS). The dataset obtained is relatively small due to the use of closed-architecture third-party LIS in UMMC, which has restricted the accessing of non-medical personnel and the direct export of the pathology reports. Therefore, the pathologist is required to copy each pathology report manually from the LIS to a Docx file. Consequently, there is an increase in the time needed and the difficulty in obtaining the pathology reports within the time constraint. However, all the datasets obtained were randomly selected to reduce bias in the datasets.

In the initial step, the reports of a patient will be extracted from the system. This report may contain related breast pathology reports of the same patient in a single document. Subsequently, the next step is to separate these reports from the same patient. Moreover, a pathology report may include one or more specimens (number of specimens in this study = 593). Figure 1 shows the composition of a patient’s pathology reports.

These reports contain three main sections: Macroscopy, microscopy, and interpretation. The macroscopy section delineates the measurement size of lesions and their margins that can be observed under naked eyes. The microscopy section describes the measurement size of lesions and their margins in the samples under the microscope. It also describes whether the cancer cells are in the lymph channels or lymph nodes. Pathologists also report the results of the ImmunoHistoChemistry (IHC) test in the microscopy section. The interpretation section describes the overall condition of the examination, such as the breast laterality, type of procedure, histologic type, and grade.

This study has been approved by the Medical Research Ethics Committee (MREC), UMMC, Kuala Lumpur (MREC ID NO: #733.22) to develop a point of care data capture for institutional breast cancer registry. The data used were de-identified secondary data from EMRs. In the development of the rule-based NLP algorithm in this study, a total of 415 pathology specimens from 174 patients (70% of the total dataset) were randomly selected for the training process, while the remaining 178 pathology specimens from 52 patients (30% of the total dataset) as testing data to evaluate the information extraction ability of the algorithm. Figure 2 shows the proposed workflow. The pathologist from the Department of Pathology, UMMC, involved in this study, identified several key data elements from the pathology report. Table 1 shows the data elements identified by the pathologist.

### 2.2. Rule-Based NLP Algorithm

The algorithm in this study was implemented in R programming language version 3.6.1. The first version of the NLP algorithm comprising a set of rules to convert the narrative report into a synoptic format was developed based on the training set. It was then improved according to the pathologist’s suggestions after verifying the results extracted manually by the pathologist. These rules were interpreted with a specific ordering, called a decision list, to resolve the ambiguity in the rule-based system.

Before the extraction process, the algorithm proposed in this study not only classifies the different reports from the same patient but also classifies each specimen from the same report, as illustrated in Figure 1, automatically by recognizing the report reference number in the document. As a result, one free-text diagnosis row was available for each specimen as an input to the NLP algorithm. Figure 3 illustrates the data input of the NLP algorithm obtained from a report.

In the first pre-processing step after reading the single diagnosis row as input, all the characters were changed to lowercase to avoid case-sensitive issues during key element extraction. Next, meaningless symbols and blank spaces included in the report that was generated when moving across platforms from LIS were removed. Roman numerals used to indicate the histologic grade value were converted to Arabic numerals to maintain consistency of format. Word stemming using the R package “hunspell” [19] was performed to reduce the variety of terms in the report.

Since the raw pathology reports were narrative, the possibility of them containing misspelled terms is high. Henceforth, this might affect the efficiency in the following extraction step. With the aim to correct any misspellings in the input data, our NLP algorithm included a spell checker function that could identify misspelled words and return the correct version by utilizing the R package “hunspell”. This package was selected because of its convenience in adding customized dictionaries to the current dictionary without overwriting the existing contents. After normalization, the specimen in the report was split into three main sections by recognizing the occurrence of labels for each section, creating semi-structured data. Successive pre-processing steps used a subset of sections or applied specific rules to different sections to greatly reduce the computation time. Each section was further subdivided into sub-rows coincidentally with every new paragraph. Then, each sub-row was analyzed using NLP rules.

When performing information extraction, several R lists were created. The structure of the list that contains a set of syntactic expressions used to match the pattern within the input text is shown in Equation (1), where the PredefinedList is an R list that stores n number of the regular expression for each specific data element.
(1)$${\mathrm{PredefinedList}}_{\mathrm{dataElement}\_a}=c\left(“\mathrm{expression}\_1”,\;“\mathrm{expression}\_n”\right),$$

The information of the type of procedure is commonly reported in the interpretation section. The algorithm matches the information within the text and the predefined list created from the interpretation section. However, in some cases, the pathologists may report the procedure type in the macroscopy section. Consequently, the algorithm analyzes all diagnosis rows in the macroscopy section only if the result was absent in the interpretation section. In the extraction of the examination date, the R package “lubridate” [20] was utilized. Integrating this package into the algorithm simplifies the extraction of date in various kinds of formats such as day (D)/month (M)/year (Y), M/D/Y, or Y/M/D, where the month can be written in a numerical or alphabetical format easily. The algorithm automatically extracts the dates from the report in the format of Y/M/D to ensure the format uniformity of the date’s value in all reports.

In the extraction of types of the lesion with its measurement size and distance to different margins from the macroscopy and microscopy section, the algorithm first draws out all the measurements with three-dimensional as a priority and followed by two-dimensional and then one-dimensional in every row of input. In this way, it avoids duplication in the extraction of measurements. The measurements of the specimens, normally in three-dimensional, are similar to the sizes of the lesions, which were reported by the pathologists in the report. However, it is less important for a clinician when reviewing a patient. Therefore, our algorithm was designed to differentiate the measurement whether it indicates a specimen or tumor by recognizing the keyword “specimen” or other terms used to describe a lesion that occurred in a sentence. In extracting the distance of the margins, it must meet the following criteria, measurement in one-dimensional (1D) or measurement within a range (R), and margin’s keywords (K) were present in the sentence as shown in Equation (2).
(2)$$\mathrm{MarginDistance}=\left\{\;\left(1D\cup^{\;}R\right)\cap^{\;}K\;\right\},$$

When extracting the presence (P) of lymphovascular invasion, skin change, and Paget disease, the sentences containing the keywords for these data elements (K) were identified, and are shown in Equation (3). Hence, negation detection was performed on the particular sentences by a set of defined linguistic rules. A list of negation words (n) used in clinical writings was created based on the training dataset to perform negation detection. Then, the location of the negation terms in the sentence was located. Different data elements may be present in the same sentence; hence, with these locations defined, the algorithm can identify which data element was negated by selecting the closet negation term to the specific keyword (N) (see Equation (4)). The absence of the targeted data elements can be defined as in Equation (5).
(3)$$P_{\mathrm{dataElement}\_x}=\left\{\;K\;\right\},$$
(4)Dist(n,K)=|n− K|, 
(5)$$\neg P_{\mathrm{dataElement}\_x}=\left\{\;K\cap^{\;}N\;\right\},\;$$

There were two grading systems used in pathology reporting, which were the NSBR grading and the DCIS grading system. The NSBR grading system uses the numeric value in grading (grade 1 to 3), while the DCIS grading system uses terms that are low, intermediate, and high in grading. This difference in grading helps our algorithm to differentiate these data elements. Most of the time, the number of lymph nodes examined and the number of lymph nodes that showed malignancy were reported together in one sentence. Hence, more semantic rules were required to identify the sentence boundary to extract the result correctly as both data elements’ responses were in the same numeric format. The algorithm draws a new sentence boundary by identifying the presence of comma punctuation or conjunction keywords.

In order to extract the histologic type of specimen, a list of morphology terms often used was predefined. When the morphology description identified is fully included in another morphology description that is matched with the predefined list, the more specific one is prioritized. For example, “papillary carcinoma” is included in a more specific description, “papillary carcinoma with invasion”. While with the presence of a decision list, the more precise description was extracted as a priority. In the ancillary studies, which included the breast biomarker ER, PR, and HER2 testing results, the terms used to describe the outcomes for biomarker ER and PR were “positive” or “negative” or a percentage of staining. While in HER2 result reporting, other than terms “positive” and “negative”, some pathologists will report it using a scoring method from score 0 to 3+. Pathologists often describe these three biomarkers in the same sentence; hence, the location of the biomarkers and the respective results are critical in getting the correct result during extraction. The distance formula (see Equation (4)) was applied to identify the test result of the targeted biomarker. Table 2 lists the examples of regular expressions used for the key data elements.

### 2.3. Synoptic Report

The synoptic report proposed in this study not only indicates the running text of different elements that are mentioned in separate lines, but it was in a more confined structure, as defined by the CAP [8]. In that way, the required data elements adhered to a paired format where each required data element is followed by a response. Therefore, separated data elements with their responses extracted from the narrative report by the NLP algorithm were displayed on separate lines.

A checklist-style pathology reporting template was also created as an alternative to create a synoptic report. Both the checklist-style pathology reporting template and the NLP algorithm shared the same database in order to maintain the consistency of data input. Since the results extracted by the NLP algorithm were in a structured format, the data can be easily integrated into the MySQL database through the R package “RMySQL” [21]. The primary key “reportID” in table report_info acts as the foreign key in the rest of the database tables to enable the linking among tables. Figure 4 illustrates the entity-relationship diagram (ERD) of the database.

### 2.4. Evaluation of the NLP Algorithm

Human validation was performed to evaluate the performance of the NLP algorithm. The pathologist in UMMC was provided with all the documents of the unannotated narrative pathology reports and the synoptic pathology reports generated automatically by the proposed algorithm, including both training and testing datasets. To validate the result of the system, the pathologist marked the extracted results as either present or absent in the narrative pathology report and correctly or incorrectly identified from the report. Borderline cases can occur when the number of specimens in each section does not match. For example, there were two examined specimens in the macroscopy section, while only one specimen was reported in the microscopy section. In this scenario, the accuracy of the extraction task may be affected. As a solution, the text mining program will add a reminder message on this problem to the initial of the program-generated synoptic report, so that the clinicians can pay more attention to the problem addressed. Precision, recall, and F1 scores were computed for each classification category in the algorithm’s evaluation performance:(6)$$\mathrm{Precision}=\frac{\mathrm{True}\;\mathrm{positive}}{\mathrm{True}\;\mathrm{positive}\;+\mathrm{False}\;\mathrm{positive}},$$
(7)$$\mathrm{Recall}=\frac{\mathrm{True}\;\mathrm{positive}}{\mathrm{True}\;\mathrm{positive}\;+\;\mathrm{False}\;\mathrm{negative}},\;$$
(8)$$F1\;\mathrm{score}=\frac{2\times \mathrm{Precision}\times \mathrm{Recall}}{\mathrm{Precision}+\mathrm{Recall}},$$

These performance scores were first computed independently for each variables and then the average value (macro-average) was taken; thus the average value was calculated by aggregate contributions of all variables (micro-average).

## 3. Results

### 3.1. NLP Algorithm to Extract Important Variables from Breast Pathology Report

Responses of 25 data elements were extracted by the proposed NLP algorithm. These include: Number of specimen in a report;Patient’s register number;Examination date;Procedure type;Breast laterality;Histologic type;Histologic grade;DCIS grade;DCIS appearance;Margin involved (macroscopically);Margin distance (macroscopically);Lesion type (macroscopically);Lesion size (macroscopically);Margin involved (microscopically);Margin distance (microscopically);Lesion type (microscopically);Lesion size (microscopically);Skin change involvement;Presence of lymphvovascular invasion;Presence of Paget’s disease;Total lymph nodes examined;Lymph nodes show malignancy;ER testing result;PR testing result;HER2 testing result.

Once the NLP algorithm was refined and optimized on the training dataset with performance scores for micro-F1 = 0.9959 and macro-F1 = 0.9931, the algorithm was then tested on the testing set data that included 178 specimens and achieved the performance scores of micro-F1 = 0.9950 and macro-F1 = 0.9897. Table 3 shows the performance score of three metrics for training data and testing data. Overall, the high F1-score in both the training and testing data shows the effectiveness of the proposed rule-based NLP algorithm in identifying the responses of the key data elements from the narrative pathology reports from a single institution, UMMC, and it was converted into a synoptic pathology report with structured data.

Other than achieving high accuracy in data element response extraction, our proposed NLP algorithm is able to split each report from the same patient and separate each specimen with the three major sections from a report correctly in all samples involved. The successful classification of each specimen helps to increase the efficiency of the proposed NLP algorithm in the following extraction step. When reporting, the unit for measurement, such as the size of the lesion or distance from a specific margin, may miss out. In Example 1–S1, the unit for the tumor’s distance from the anterior margin was missing. Hence, this caused the algorithm to recognize the numeric value as a float number and exclude it during extraction. Consequently, our algorithm included a set of rules that can differentiate the regular integer and measurement to add the missing unit. For example, unit “cm” will be added to Example 1–S1, but not in S2.

Integrating the R package “lubridate” in the algorithm helped to differentiate the date value with other numeric values such as size, distance, and patient’s register number. However, in some rare cases, the package “lubridate” extracts the date value incorrectly. For example, the date is written as “5.6.19”, the month value can be 5 or 6 depending on the decision list, concerning whether to check the month or day first. As a result, this increased the false positive number and reduced the precision score. In extracting the results (positive or negative) for the biomarkers test, the proposed algorithm can identify the value correctly even when both text “positive” and “negative” appeared in the same sentence that is shown in Example 2. The original text was “nuclear positivity” before word stemming. Other than that, our algorithm can also classify the correct result for different biomarkers even when they are mentioned in the same sentence (Example 3). This is achieved by calculating the distance between the biomarker’s keyword with results in the text.

When extracting the lesion size measurement, the sizes of specimens (Example 4–S1) were excluded in the extraction process even when it had the same format as lesion size (Example 4–S2). Our algorithm differentiated the size measurement by recognizing the keyword “specimen” and other keywords that describe a lesion. Different lesion sizes (Example 4–S2 and S4) and margin distance sizes (Example 4–S3 and S5) could extract and store separately in the database table to ensure correctness when generating the synoptic report. However, there are still some false positive cases which are shown in Example 5. The algorithm extracted the type of lesion involved as “cystic spaces”, while the exact response should be “fibrosis”. This is due to both morphology terms being presented in the predefined list, and the algorithm will only select the nearest keyword to the measurement.

In some cases, the reporting can be ambiguous, such as the information was written by pathologists and had a mismatch in different sections. In Example 6, the number of lymph nodes stated in the microscopy section is “6”, while in the interpretation section, the number reported is “2”. This leads to an increase in the false positive rate. The algorithm failed to extract the appearance of DCIS in Example 7 since the keyword “DCIS” is not present in the sentence; hence, it was skipped by the algorithm and increased the false negative value. In some rare cases, the lesion involved in the margin distance was not mentioned by the pathologists but was reported in different sections (Example 8). Hence, the algorithm could not extract the type of lesion involved in the specific margin, leading to an increase in the false negative rate.

The examples of diagnosis row existing in the pathology report are:

**Example** **1.**
***S1*** 
*“The tumor is 1 from the anterior margin and 1.5 cm from the deep margin.”*
***S2*** 
*“Block 7—medial margin (shaved).”*



**Example** **2.**
*“ER: Negative, <1% of nuclear positive.”*


**Example** **3.**
*“The tumour cells show strong nuclear positivity (>90%) for ER and PR and negative for HER2 (0–1+).”*


**Example** **4.**
***S1*** 
*“A mastectomy specimen weighing 350 g, measuring 18 cm × 13 cm × 3 cm.”*
***S2*** 
*“Serial cut sections show a well circumscribed tumour 1.7 cm × 1.5 cm × 0.6 cm.”*
***S3*** 
*“It is abutting the deep margin which is 0.1 cm away, 1.5 cm from the superior margin, 3.6 cm from inferior margin, 0.1 cm from the medial margin and 17.5 cm from the lateral margin.”*
***S4*** 
*“There is an area of firm whitish comedo-like lesion 3.5 cm × 2.5 cm × 2 cm.”*
***S5*** 
*“It is 0.5 cm from the deep margin, 0.1 cm from inferior margin, 3.5 cm from superior margin, 2 cm from medial margin and 10 cm from lateral margin.”*



**Example** **5.**
*“Shows scattered areas of fibrosis with a focal area displaying cystic spaces measuring 0.3 cm in maximum dimension.”*


**Example** **6.**
*In microscopy section: “A total of 6 reactive lymph nodes present which shows no tumor involvement.”*

*In interpretation section: “2 lymph nodes. No tumour metastasis”*


**Example** **7.**
*“The neoplastic cells are arranged in a solid pattern with foci of comedo-necrosis.”*


**Example** **8.**
*“The nearest margin is the deep margin at 1 mm away.”*


### 3.2. Synoptic Report

After the extraction step, the extracted values were matched to their data elements as a “response”; each data element with its corresponding response was displayed on separate lines. Table 4 illustrated some examples of converting narrative information in pathology reports to a synoptic format. A checklist-style reporting template (see Appendix A) that shared the same database was created to maintain the data structure’s consistency concerning existing and newly inserted reports.

## 4. Discussion

The rising data in oncological diagnostics increased the complexity of pathology reports. However, the pathology reports from LIS are reported as a free text with different sections instead of as distinct categories of each data element. Clinicians are needed to review and interpret the key point from the long narrative report to determine the following treatment process. Most of the clinical information was stored in textual form, and the database structure was different for structured data. In such a scenario, NLP offers an opportunity to automate the encoding of narrative reports into clean and structured data, hence producing a synoptic report as an alternative to the costly manual data extraction process [22,23]. However, the majority of the NLP techniques applied in the clinical domain mainly focus on the primary tasks, such as ureteric stones identification [24], stroke detection [25], generating section label [26], or cancer morphology classification [27], or they focus on classification tasks involving fields with a few labels, for example, site, laterality, behavior, histology, and grade [6].

We developed an automated rule-based NLP to extract responses for 25 data elements from the pathology reports, thus converting them into a synoptic pathology report. These reports were characterized by a high variability with the different writing styles and a highly unstructured nature to label the same data element in the different pathology reports written by various pathologists. Henceforth, the closed-architecture third party information system used in the institution caused the export of the existing pathology report across platforms to become troublesome. A single exported document contained different pathology reports of the same patient. So, we designed our algorithm to allow the automatic separation of each pathology report before performing the extraction process to minimize the human effort.

The proposed algorithm was developed with the aim to be used and integrated into the current reporting pipeline in UMMC to produce synoptic reports to ease the clinician’s audit and research purpose. For the purpose of achieving this objective, it was necessary to reach the highest possible micro-average performance scores (best value at 1, and worst score at 0) [28]. Therefore, our proposed automated NLP algorithm achieved encouraging results in both the training and the testing dataset when compared to the manual extraction by the expert pathologist (micro-F1 equal to 0.9950 on the test set), with the data elements, microscopy margin distance which has the highest number of false positive and false negative values, 2 out of 27 and 3 out of 27, respectively. These errors are mainly due to the confusing writing format, such as combining different specimens into the same section or reporting a data element in multiple sections by some pathologies (often not present in most pathology reports). A similar work based on a self-supervised convolutional neural network-based algorithm was developed by Spandorfer et al. to convert unstructured narrative computed tomography pulmonary angiography (CTPA) reports into structured reports [26]. While their algorithm promised a high accuracy, achieving 91.6% and 95.9% using strict and modified criteria, it provided only the most basic structure by applying section labels to the sentences.

Despite the rise of machine learning techniques applied in NLP, the rule-based NLP algorithm is still widely applied in clinical NLP, which is considerably different from the general NLP community. Regarding the straightforward characteristic of the rule-based algorithm, it eases the debug process by the developer through interactive refinement with the clinician’s feedback [29]. Besides that, machine learning-based NLP methods were mainly used for data prediction, estimation, and association mining. Furthermore, the difficulty in interpreting and correcting specific errors reported by the end-user due to the black box mechanism in machine learning algorithms consolidated the popularity of rule-based NLP in the clinical domain.

In the study by Hammami et al., an automated cancer morphology based on a rule-based NLP approach was developed [27]. Their algorithm achieved a successful result with a micro-F1 score of 98.14% on a single task, extracting cancer morphology codes as defined in the Third Edition of International Classification of Diseases for Oncology (ICD-O-3) from Italian pathology reports. Another rule-based NLP algorithm developed by Odisho et al. achieved an overall accuracy of over 94%. However, the data extraction from the prostate pathology report only involved a small field of data elements, such as Gleason Score, margin status, extracapsular extension, seminal vesicle invasion, and TN stage with a small range of possible values [30]. In addition, Bozkurt et al. developed three types of NLP algorithms, which were rule based, deep learning, and hybrid model, to compare the performance in classifying the severity of a prostate cancer patient using clinical notes. Their accuracy results show that the rule-based model achieved the highest accuracy of 0.86, which outperformed the deep model with an accuracy of 0.73 and a hybrid model that combined both methods with an accuracy of 0.75 [31].

Even though the generation of rules in the rule-based NLP requires human effort, its transparent characteristic is essential for clinical application to incorporate domain knowledge from knowledge bases or experts. Most machine learning-based techniques require an enormous set of well-curated input to promise accuracy in a specific task, such as data prediction, classification, and association mining. For example, in the study conducted by Levy et al., they developed a current procedural terminology code predictor by support vector machine (SVM), extreme gradient boosting (XGBoost), and bidirectional encoder representations from transformer (BERT) from 93,039 pathology reports [32]. In another study by Kalra et al., four types of machine learning-based NLP algorithms—linear SVM, radial basis function SVM, logistic regression, and extreme gradient boost—were developed to classify 1949 manually cleaned pathology reports into different diagnosis categories [7].

Synoptic reporting, an alternative reporting style, has shown a significant increase in the completeness of data elements in pathology reports across various diseases, including but not limited to breast, upper gastrointestinal, lung, colon, and prostate cancers [33,34,35,36,37]. Other than these favorable quantitative outcomes, Yunker’s study showed that synoptic reports reduce time spent on the production of the report by pathologists [38]. Furthermore, increasing the convenience of reading the diagnosis report helps to improve its quality. In several studies, the use of standardized proforma in colorectal cancer pathology reporting greatly improved the quality of the report by improving the mean number of lymph nodes identified in the surgical specimen [36,39,40]. Other than that, the highly structured synoptic reports are more amenable and valued for secondary use, as most of the analysis models are derived from structured data [4].

Nevertheless, Lankshear’s study drew an opposite conclusion in the time required to produce a cancer pathology report. Lankshear’s study showed that the pathologists who were provided with a five-point Likert scale (1 = significantly less than narrative reports, 3 = about the same, and 5 = significantly more than narrative reports) reported that slightly more time is required (mean score = 3.51) which indicated the synoptic report required 25% to 50% more time to reach completion. However, the majority physician group (60%) reported that the time required to obtain the final pathology report in synoptic format was about the same as the narrative format [9]. The length of the reports can be another issue for synoptic reports [5]. In narrative reporting, it is acceptable to omit the absent data element. While in synoptic reporting, the absent data element will be reported as “not applicable” and hence increase the length of reports. Overall, the advantages of synoptic reports outweigh the disadvantages, notwithstanding synoptic reports still have not been widely adopted in the clinical domain. The most critical barriers in implementing synoptic reporting are the pathologists’ personal preference for the flexibility of narrative report [41,42]. Another factor preventing the successful implementation is the incompatibility of a new reporting format in existing work environments, such as the database structure used to store the EMRs [3,43].

There are several limitations to this study. First, the proposed rule-based NLP algorithm in this study achieved more than 95% accuracy for most data elements, but part of this accuracy is dependent on the underlying patterns in the pathology reports, which are mostly standardized at UMMC. In other words, there is still room for improvement in the generalizability of the algorithm. Henceforth, these rules in NLP may require manual updating as only a small dataset was used. It involved 593 specimens in the algorithm development, and may not cover all the scenarios, hence increasing the false negative value. Currently, acquiring the pathology report is done manually by the pathologist, which can be too laborious. Therefore, future work on minimizing the human intervention on automatically obtaining and annotating the pathology report from the LIS in UMMC is needed. In addition, expanding the dataset from different institutions will be considered to increase the generalizability and further reduce the bias of the algorithm.

Our study showed significant progress in promoting the implementation of synoptic reporting in the clinical domain by presenting an automated way to convert the existing narrative report into a synoptic report to suit the working environment. This innovation is not only to digitize old reports for audit and research purposes but also prospective digitized data collection.

## 5. Conclusions

This study demonstrates a novel NLP algorithm in extracting a larger field of data elements with its respective responses that rely on ad hoc linguistic rules defined on 593 specimens from 174 patients, achieving a micro-F1 score of more than 98% in both training and testing set. A synoptic report that is highly structured promises the diverse requirements of various corresponding users. Nevertheless, the involvement of experts in different areas such as clinicians, pathologists, and data scientists is important as their domain knowledge and insight in recognizing which features work best to improve synoptic reporting for all stakeholders. In summary, our algorithm assists clinicians in highlighting the key points from narrative reports and serves as a preliminary step in promoting synoptic reporting in the clinical domain.

## Figures and Tables

**Figure 1 diagnostics-12-00879-f001:**
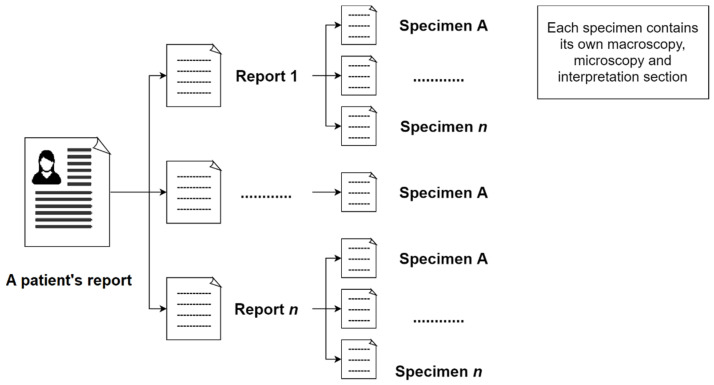
Representation of pathology report’s composition.

**Figure 2 diagnostics-12-00879-f002:**
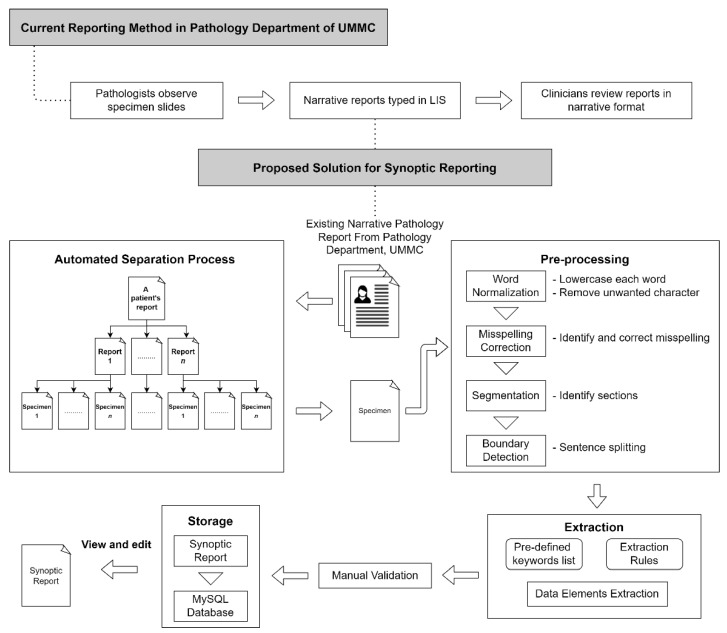
Overview of study.

**Figure 3 diagnostics-12-00879-f003:**
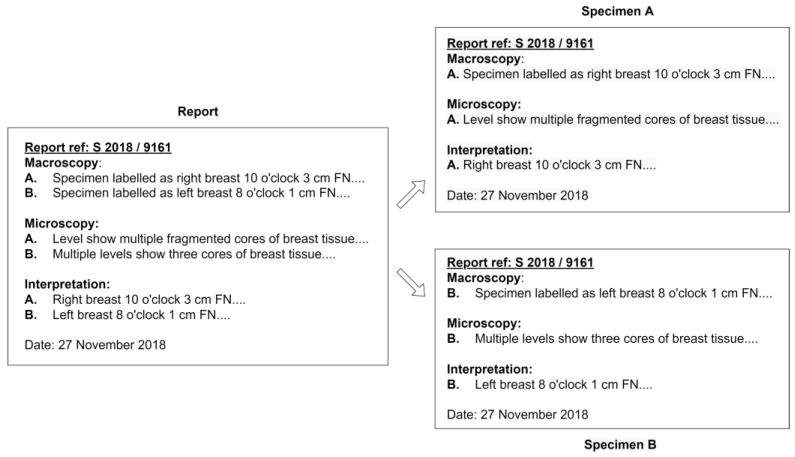
Input for the NLP algorithm.

**Figure 4 diagnostics-12-00879-f004:**
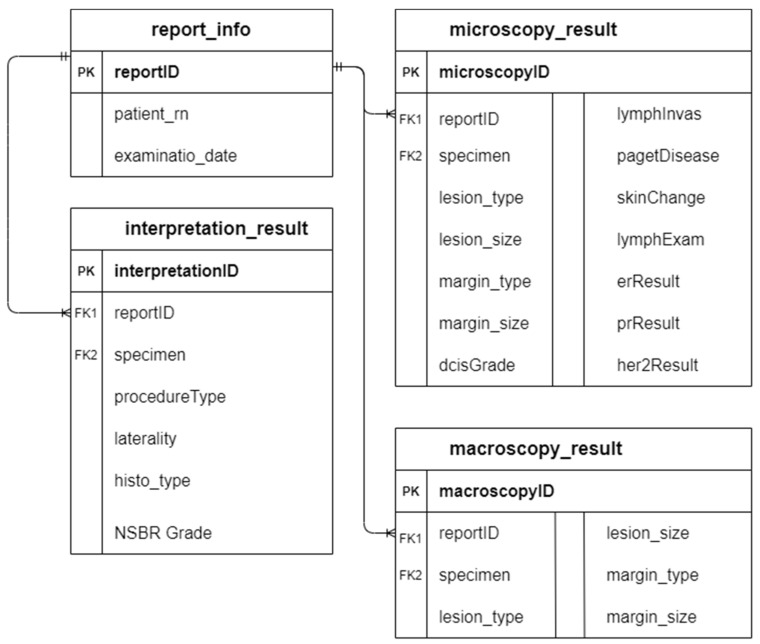
ERD of database for pathology reporting.

**Table 1 diagnostics-12-00879-t001:** Key elements identified by the pathologist in UMMC.

Data Elements	Description
**Named entity**	
Date	- Examination date.
Patient’s register number	- Unique ID for a patient.
Report’s reference number	- Unique ID for a report.
**Interpretation section**	
Type of procedure	- Procedure used to obtain specimen.
Specimen laterality	- Site of breast involved.
Histologic type	- Description of a tumor look under a microscope.
Histologic grade	- Nottingham modification of Bloom-Richardson (NSBR) grading system based on tumor tubule formation, number of mitotic figures in most active areas and nuclear pleomorphism.
**Macro and microscopy section**	
Lesion	- Type of lesions and its size observed macro and microscopically.
Margins	- Distance of lesion from different margins.
**Microscopy section**	
Ductal Carcinoma In Situ (DCIS) grades	- Grading to describe the distance cancer cells resemble normal breast cells and how fast they grow.
DCIS appearances	- Architectural growth pattern of DCIS.
Lymphovascular invasion	- Presence of tumor cells in lymphatics or blood vessels.
Skin change involvement	- Presence of skin change include puckering, dimpling, a rash, or redness of the skin of the breast.
Paget disease	- Presence of eczema-like changes to the skin of the nipple and the area of darker skin surrounding the nipple.
Regional lymph nodes	- Number of lymph nodes examined and number of lymph nodes involved by tumor cell.
Ancillary studies	- Breast biomarker testing results for estrogen receptor (ER), progesterone receptor (PR) and human epidermal growth factor receptor 2 (HER2) by IHC.

**Table 2 diagnostics-12-00879-t002:** Variables extracted with its regular expression.

Data Elements	Regular Expression
Type of procedure	- Needle biopsy; hook-wire localization biopsy; excision; mastectomy
Type of lesion	- Tumor; fibrotic lesion; fibrosis; cyst; mass; nodule
Size of lesion	- [numeric][units]? × [numeric][units]? × [numeric][units]; [numeric][units]? × [numeric][units]; [numeric][units]? * where symbol “?” indicated that the value is optional
Margins distance	- [numeric][units]; <[numeric][units]; >[numeric][units]; [numeric][units]? − [numeric][units]
Margins involved	- Anterior; deep; superior; inferior; medial; lateral; posterior; superficial; peripheral; axis
Presence of lymphovascular invasion	- Lymphovascular invasion; Lymphovascular permeation Terms to described if present: - Seen; noted; presented; observed; detected Terms to described if absent: - No; not detected; absent
Skin change involvement	- Skin change; skin lesion; skin Terms to described if present: - Changes are seen; seen; noted; presented; observed; detected Terms to described if absent: - No; not detected; absent
Presence of Paget’s disease	- Paget disease; Paget cell; Pagetoid spread Terms to described if present: - Seen; noted; presented; observed; detected Terms to described if absent: - No; not detected; absent
NSBR grade	- Grade 1; grade 2; grade 3
DCIS grade	- Low; intermediate; low to intermediate; high
DCIS appearance	- Cribriform; micropapillary; papillary; solid; flat or clinging; comedo
Histologic type	- No residual invasive carcinoma; Invasive lobular carcinoma; Invasive cribriform carcinoma; papillary carcinoma with invasion; papillary carcinoma
Total number of lymph nodes examined	- identified [numeric] lymph node(s); all [numeric] lymph node(s); identified a total of [numeric] lymph node(s); [numeric] out of the [numeric] lymph nodes examined
Number of lymph nodes show malignancy	- metastatic carcinoma in [numeric] out of the [numeric] lymph nodes; [numeric] lymph nodes are effaced and replaced by malignant cells|
ER test result	- Positive; negative; weak staining; strong staining; less than [numeric]% staining; more than [numeric]% staining
PR test result	- Positive; negative; weak staining; strong staining; less than [numeric]% staining; more than [numeric]% staining
HER2 test result	- Positive; negative; equivocal; overexpressed; not overexpressed; score 0 to 3+

**Table 3 diagnostics-12-00879-t003:** Evaluation result of training and testing data.

		Precision	Recall	F1 Score
**Training set**	**Micro-averaged**	0.9958	0.9960	0.9959
	**Macro-averaged**	0.9926	0.9936	0.9931
**Testing set**	**Micro-averaged**	0.9942	0.9959	0.9950
	**Macro-averaged**	0.9820	0.9914	0.9897

**Table 4 diagnostics-12-00879-t004:** Examples of pathology report information in narrative and synoptic format, respectively.

Narrative	Synoptic
- Received a mastectomy specimen weighing 790 g and measuring 17.5 cm × 15.5 cm × 4.5 cm.	- Procedure: mastectomy
- Specimen labelled as right breast retro-areolar.	- Specimen laterality: right
- Interpretation: Invasive carcinoma, nst	- Histologic type: invasive carcinoma of no special type
- Interpretation: Bloom-richardson grade 2	- Histologic grade: Grade 2 (Bloom and Richardson)
- There is no lymphovascular invasion	- Lymphovascular invasion: absent
- They are focally positive for PR and ER. The HER-2 expression is negative (0).	- Ancillary studies: ER biomarker result: positive PR biomarker result: positive HER2 biomarker result: negative, score 0
- Sections show a total of 10 lymph nodes with no evidence of tumour metastasis.	- Regional lymph nodes: Lymph node(s) examined: 10 Lymph node(s) show(s) malignancy: 0
- A small bit of skin tissue is seen and no Paget’s disease is observed.	- Paget’s disease: absent
- The specimen measures 13.5 cm × 14.5 cm × 4 cm, a serial section shows a tumour bed measures 3 cm × 2 cm × 1.5 cm located at upper outer quadrant.	- Lesion size (macroscopy): Tumor bed size: 3 cm × 2 cm × 1.5 cm
- It is 3.5 cm from superior margin, 7.5 cm from inferior and medial margins, 4.5 cm from lateral margin, 0.5 cm from anterior margin and 1 cm from deep margin.	- Margin (macroscopy): Tumor Bed margin: superior margin = 3.5 cm, inferior margin = 7.5 cm, medial margin = 7.5 cm, lateral margin = 4.5 cm, anterior margin = 0.5 cm, deep margin = 1 cm

## Data Availability

Breast pathology report data used in this study cannot be shared publicly because it consists of personal information. Hence full data access through a public repository is not permitted by the institution. Any enquiries about data used, please contact sarinder@um.edu.my.

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
