# Peer review of "Automated Generation of Synoptic Reports from Narrative Pathology Reports in University Malaya Medical Centre Using Natural Language Processing"

_diagnostics, 2022, doi:10.3390/diagnostics12040879_

Round 1

Reviewer 1 Report

Overall

This research article introduces a system that generates synoptic reports through a rule-based NLP pipeline from narrative pathology reports. This structuring of the data enables the extracted data to be more easily processed, saving valuable time for medical staff. The introduction clearly introduces the problem that needs to be addressed and shows how this study fills that research space. However, there is only a cursory mention of natural language processing, with just three citations to related work with no citations to very recent work (i.e. last three years). This makes it impossible to describe the literature view as state of the art. The method and choices made for the study are appropriate and the rationale for choices is provided when necessary.  The NLP procedure selected is fairly standard for data extraction. What makes this paper novel is the application to this particular genre of writing. The results and discussion are well structured and logical. The discussion of the strengths and weaknesses of the approach are appropriate, and should help other researchers build on this study. The tables and figures included are readable, appropriate, and referred to in the body of the paper. This study will be of interest to the readership. I have a few major suggestions, and a number of minor suggestions to improve the clarity of the paper.

Major changes

  1. Extend the literature review for NLP in medical contexts and natural language generation (ideally in medical texts).
  2. The data set is rather small for typical NLP studies. Please explain why such a small sample was selected; and, more importantly, why its size is not an issue. The downside to the dataset size is mentioned in the limitations, but a justification for the size selected is needed.
  3. The validation procedure is rather vague. Was human validation used for a subset or the whole set of training/test data? Typically, in classification tasks, borderline cases occur. Please describe any such cases and how they were resolved.

Minor changes

I have provided line numbers for each suggestion to speed up the revision process. Feel free to use alternative wordings to those I suggest.

Lines 4 

The final word of the title  “Technique” is redundant. 

Lines 13 – 15    

The second sentence of the abstract does not make sense. I cannot understand how the final prepositional phrase “by making narrative reporting necessary” is linked to being responsible for training.

Line 25 

This achievement had correlated to - - > This achievement correlated to

Line 95

Pathologists also reports - - > Pathologists also report

Line 107 Figure

Extaction - - > Extraction

Line 113

that made up of  - - > made up of OR that was made up of OR comprising

Line 115

improvised - - > Perhaps, you mean improved?

Line 198

The different elements’ running text - - > the running text of different elements (easier for readers to understand)

Line 289

Differential - - > differentiated OR a more suitable verb

Line 295

This is due to both morphology terms were presented - - > This is due to both morphology terms being presented

Author Response

Pleasa see the attachment.

Reviewer 2 Report

The introduction must clearly present the objectives of the paper.

The description of similar approaches in the literature (briefly illustrated in the introduction) needs to be improved.

These two papers can be suggested:

Nicholas A. I. Omoregbe, Israel O. Ndaman, Sanjay Misra, Olusola O. Abayomi-Alli, Robertas Damaševičius, "Text Messaging-Based Medical Diagnosis Using Natural Language Processing and Fuzzy Logic", Journal of Healthcare Engineering, vol. 2020, Article ID 8839524, 14 pages, 2020. https://doi.org/10.1155/2020/8839524

V. Carchiolo, A. Longheu, G. Reitano and L. Zagarella, "Medical prescription classification: a NLP-based approach," 2019 Federated Conference on Computer Science and Information Systems (FedCSIS), 2019, pp. 605-609, doi: 10.15439/2019F197.

Table 1 and 2 must be restructured to make the information more easily readable.

The algorithm (section 2.2) should be introduced in a more formal way and then described qualitatively.

Round 2

Reviewer 1 Report

Overall

The authors have addressed the major issues that I pointed out in the initial review and have corrected all of the minor issues. Apart from a very minor change to one of the new references, I have no more objections to the publication of this paper.

Major changes

I have no additional major changes to suggest

Minor change

I have only one new suggestion.

Line 581 Reference 16

The cited journal does not provide page numbers, but provides article IDs in lieu. Please add the article ID number – 8839524.

Author Response

Minor changes
Point 1:
Line 581 Reference 16
The cited journal does not provide page numbers, but provides article IDs in lieu. Please add the article ID number – 8839524.

Response 1:
Correction on the citation has been made.